# HuWo: Building Physical Interaction World Models for Humanoid Robot Locomotion

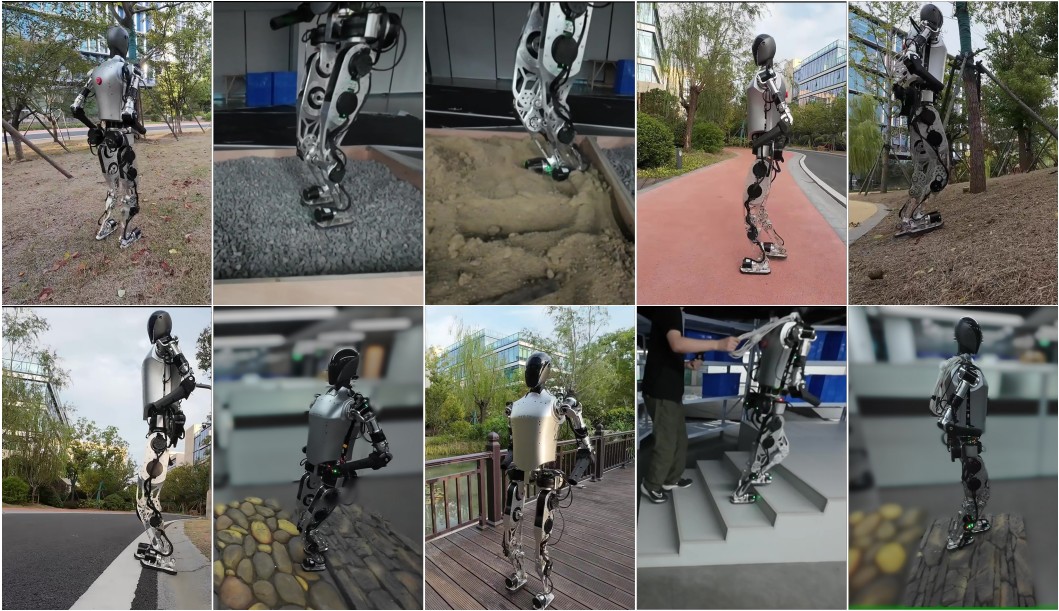

Figure 1: The proposed method enables stable humanoid robot walking without external sensors. Experiments in diverse environments—grassy fields, rocky ground, sandy surfaces, terrains with varying hardness and friction (soft rubber, smooth wood, hard asphalt), five-level stairs, and artificially constructed complex terrains—demonstrate its adaptabili.

## ABSTRACT

Reinforcement learning has proven effective for humanoid robot locomotion, yet achieving stable movement in complex environments remains challenging. Humanoid robots must maintain balance while navigating and continuously adapt to interactions with the environment. A deep understanding of these robot-environment dynamics is essential for achieving stable locomotion. Since there is privileged information that the robot cannot directly access, to expand the observable space, previous reinforcement learning-based methods either reconstruct environmental information from partial observations or reconstruct robotic dynamics information from partial observations, but they fall short of fully capturing the dynamics of robot-environment interactions. In this work, we propose an end-to-end reinforcement learning control framework based on physical interaction **Wo**rld Model for **Hu**manoid Robots (HuWo). Our primary innovation is the introduction of a physical interaction world model to understand the dynamic interactions between the robot and the environment. Additionally, to address the temporal and dynamic nature of these interactions, we employ the hidden layers of Transformer-XL for implicit modeling. The proposed framework can showcase robust and flexible locomotion ability in complex environments such as slopes, stairs, and discontinuous surfaces. We validated the robustness of this method using the $Zerith1$ robot, both in simulations and real-world deployments, and quantitatively compared our HuWo against the baselines with better traversability and command-tracking.

## 1 INTRODUCTION

Humanoid robots are expected to perform tasks related to human activities and work alongside humans, which includes possessing motion capabilities comparable to humans and adapting their gaits to various terrain conditions. Although they exhibit superior mobility compared to wheeled robots in complex terrains, controlling them in scenarios with discontinuous contact and diverse motion skills remains challenging. Transitioning natural movements to humanoid robots still faces long-term technical challenges, including but not limited to the high degrees of freedom, underactuation, and complex non-linear dynamics of humanoid robots.

Traditional model-based control methods have significantly enhanced the locomotion capabilities of humanoid robots by using physical models to predict robot behavior. Wensing & Orin (2014); Chignoli et al. (2021); Ahn (2023) However, these methods rely on accurate environmental dynamics modeling, which limits their application in complex terrains. Simplified dynamic models often lead to conservative movements, restricting the robot's potential. In contrast, reinforcement learning-based methods do not rely on detailed physical modeling and have shown greater flexibility and adaptability on legged robots. However, for humanoid robots, these methods can only handle relatively simple environments and have not yet fully addressed dynamic control issues in complex terrains.

Environmental information and robot motion information are essentially information from different domains, and result in understanding their interactions is challenging. Since actor networks can only obtain partial observations of the environment, they generally reconstruct partial observations into more complete environmental information by incorporating historical information or additional observational data. While these methods can reconstruct environment or robot dynamics information from partial observations, they fail to fully characterize the physical interaction processes between the robot and the environment. To address this issue, we introduce Building dynamical Interaction World Models, which employ self-attention mechanisms to learn compact representations of historical observation inputs and implicitly infer latent interaction states by predicting future observation states. This approach better captures the complex interaction processes between the robot and the environment, enhancing the environment understanding and enabling stable locomotion on complex terrains.

Our input consists of temporally related historical sequence information, and we use the Transformer-XL Dai (2019), which allows the world model to directly access observations from previous time steps and learn long-term dependencies. The Transformer structure comprises multiple residual connections and self-attention mechanisms. The self-attention mechanism has unique advantages in modeling sequential information because it captures global information in the sequence without relying on fixed time windows.

We demonstrate the entire framework on the affordance-based bipedal platform $Zerith1$ and validate our method. With our approach, the robot can traverse complex terrain in both simulation and the real world. Overall, our main contributions are:

- We propose a physical interaction world model, representing the first application of Transformer-XL based world model framework to humanoid robot tasks. By integrating it with the actor-critic method, we achieve enhanced reinforcement learning exploration capabilities.

- Our approach incorporates time series information into the critic and leverages the world model for future predictions, significantly improving the critic network's ability to evaluate the robot's state and facilitating more globally informed decision-making.

- Our method bridges the gap between simulation and the real world. Both simulation and real-world experiments demonstrate its superior traversability and command-tracking performance, fully showcasing the robustness of the approach.

## 2 RELATED WORK

**Blind Legged locomotion** For legged robot locomotion control, model-based methods are often difficult to generalize in an environment that is not modeled. Meanwhile, imitation learning Escontrela

et al. (2022); Luo et al. (2023); Radosavovic et al. (2024b) needs to rely on reference motion trajectories, but morphology and mass difference between human and robots result in scarce valid data. In contrast, reinforcement learning(RL) can not only generalize to new environments, but also does not rely on reference trajectories. However, RL control also faces the challenge of Sim2Real Gap and limitation of perception, to solve this problem, there are a number of approaches Lee et al. (2020); Kumar et al. (2021); Lai et al. (2023); Wei et al. (2023)that utilize teacher-student strategy, with the teacher model receiving complete information. The output of the teacher model is then used to supervise the student model. In order to be able to better estimate privileged information that cannot be observed, some methods feed richer information such as gait Margolis & Agrawal (2023); Li et al. (2024); Castillo et al. (2023)the controller, and some methods introduce state estimator modules Ji et al. (2022); Nahrendra et al. (2023); Long et al. (2023), compensating for partial observability by expanding the state space. Our approach is also intended to enrich the observation space. However, by integrating a world model, we can better understand the deeper information embedded in the current observations—specifically, the interaction between the robot and its environment—through predictions of future observations.

**World model for humanoid** The initial world model Ha & Schmidhuber (2018) is inspired by how humans process complex information to form an abstract representation of the world, understanding key entities and their interactions, and creating an internal representation of the world that allows predicting future events and making quick responses. For various problems that can be addressed using reinforcement learning, the Dreamer series algorithms Wu et al. (2023); Hafner et al. (2020; 2023) have systematically explored the construction and learning of world models as well as the optimization of value and policy functions based on the actor-critic paradigm. Daydreamer Wu et al. (2023) employs online learning, focusing on predicting future outcomes through experience with the world model and using these predictions to reduce the trial-and-error process in the actual environment, thereby improving training efficiency. The world denoising model Gu et al. (2024) addresses the issue of discrepancies between simulation and real-world environments by utilizing the predictive capability of the world model for denoising. However, unlike the aforementioned methods, we innovatively apply the denoising model to abstract implicit features of the dynamical interaction between the robot and the environment, leveraging these features for decision-making and enabling robust locomotion in humanoid robots.

**Transformers for humanoid** The Transformer Vaswani et al. (2017) excels in handling long sequences and is compatible with various modalities and their combinations. It has achieved remarkable results in fields such as vision Dosovitskiy et al. (2020); Arnab et al. (2021); Touvron et al. (2021) and natural language processing Devlin et al. (2018); Radford et al. (2018; 2019); Brown (2020). In reinforcement learning, decision-making methods such as Trajectory Transformer Giuliari et al. (2021) and Decision Transformer Chen et al. (2021) have been developed. For legged robot motion control tasks, Lai et al. (2023) successfully deployed a control strategy to a quadrupedal robot by leveraging Decision Transformer and a two-stage knowledge distillation approach. Yang et al. (2021) trained RL algorithms with a high-level vision controller to process visual and proprioceptive information and output target linear and angular velocities for driving lower-level controllers. Fu et al. (2024); Radosavovic et al. (2024a) used Transformers as feature extractors to achieve simple walking for humanoid robots. However, a common challenge in control tasks is that Transformers cannot capture the relationships between different segments, whereas our method using Transformer-XL establishes connections between different segments, avoiding information fragmentation.

Methods such as Chen et al. (2022); Micheli et al. (2022); Robine et al. (2023); Zhang et al. (2024); Deng et al. (2024) combine Transformers with world models. Through this integration, We introduce a novel Humanoid Locomotion Framework **HuWo**, in our method, transformers enable the world model to access past state information directly, rather than relying on compressed information, thus reducing the data compression process.

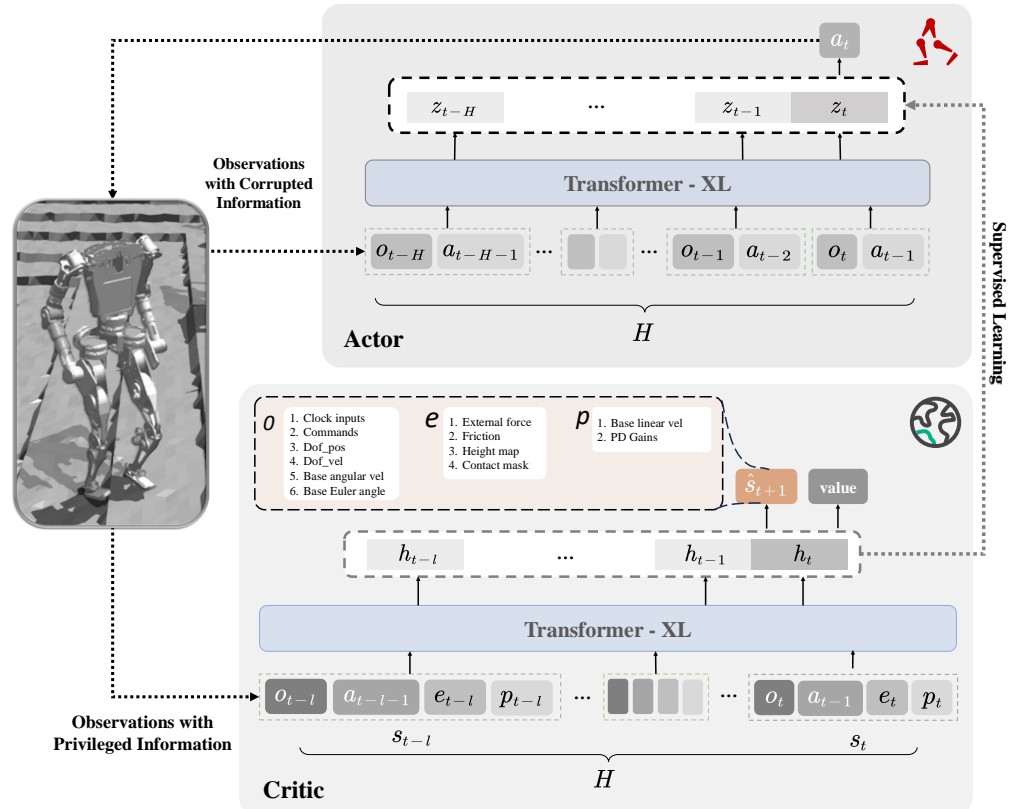

Figure 2: Overview of **HuWo**. The framework consists of an actor and a critic network implemented based on transformers. The actor network processes observation with corrupted information $o_t^H$, while the critic network processes observation with privileged information $s_t^H$ to predict the next observation $\hat{s_{t+1}}$ through the dynamic model. The hidden variables $h_t^H$ obtained by the critic supervises the learning of the actor's hidden variables $z_t^H$, allowing the actor policy to effectively learn interaction information via a regression network.

# 3 METHOD

## 3.1 PRELIMINARY

### 3.1.1 REINFORCE LEARNING TASK

In this paper, we formulate the problem of humanoid locomotion in complex terrain as a partially observable Markov decision process (POMDP) with discrete time steps $t \in N$, $\mathcal{M} = [\mathcal{S}, \mathcal{O}, \mathcal{A}, T, Z, r, \gamma]$, where $S$, $\mathcal{O}$ and $\mathcal{A}$ are the state, observations and action spaces. The state transition probability $T(o, a, s')$ represents the probability of receiving observation $o$ after executing action $a$ and transitioning to a new state $s'$, the observation probability $Z(s', a, o)$ represents the probability after executing action $a$ and transitioning to a new state $s'$, the reward function $R(s, a, s')$ represents the reward obtained after executing action $a$ in state $s$ and transitioning to state $s'$, and the discount factor $\gamma$ is a value between 0 and 1, which is used to weigh the relative importance of immediate rewards and future rewards. The ultimate goal is to find a strategy that maximizes the discounted reward $J(\pi) = \mathbb{E}_\pi \left[ \sum_{t=0}^{\infty} \gamma^t r_t \right]$.

### 3.1.2 TASK DESCRIPTION

In our Physical Interaction World Models, we decompose the locomotion task in complex environment into the following processes:

- **Dynamical Environment Understanding** In complex environmental locomotion tasks, robot's understanding of its physical interactions with the environment determines its sub-

sequent decisions. This process is highly dynamic and strongly temporally correlated, encompassing the relationship between environmental information and the robot's dynamic data, as well as the memory of these two types of information over historical time series and the robot's perception of environmental changes. Developing the robot's cognitive ability to interact physically with a dynamic environment is crucial.

- **Dynamic Prediction** Dynamic prediction plays a pivotal role in enabling the robot to interact effectively with its environment. By leveraging interaction information, the robot can "imagine" the complete states of the physical world and itself that would result from each possible action. This capability allows the robot to anticipate future dynamics and evaluate the potential value of its actions in a proactive manner. Such dynamic estimation not only enhances the robot's adaptability to diverse scenarios but also strengthens the generalization of its walking capabilities, particularly in complex and unpredictable environments.

- **Observation Space Expansion** The robot can only access partially observable states of the environment. However, partial observations can only capture local information and fail to comprehensively represent the full complexity of the environment, making them insufficient to support the robot's decision-making requirements in complex environments. To learn comprehensive information, the policy network needs to expand the observation space based on historical observation sequences and dynamic predictions, ensuring that each extended state provides sufficient information to compensate for partial observability.

## 3.2 PHYSICAL INTERACTION WORLD MODELS

### 3.2.1 OVERVIEW

Our proposed Physical Interaction World Models method includes a dynamics model and a physical interaction regression model. We adopt an asymmetric actor-critic architecture, where the critic network combines with the dynamics model. The critic takes the historical observation state information $s_t^H$ as input and compresses it into a hidden variable sequence $h_t^H$. The dynamics model predicts future state information $\hat{s}_{t+1}$, while the critic estimates the state value function. The actor network takes partial historical observation information $[o_t, a_{t-1}]^H$ as input and compresses it into a hidden variable sequence $z_t^H$. The actor hidden variables are supervised by the critic hidden variables to learn more interaction information. The actor network relies on the value estimation provided by the critic network to make decisions. The Transformer-XL architecture allows the world model to directly access historical observation information rather than compressed information. Due to its recursive structure, each time step and previous hidden states together determine the current hidden state.

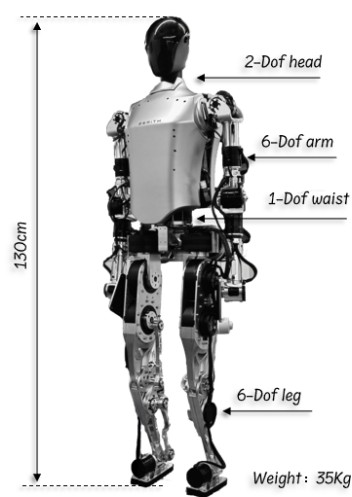

Figure 3: Overview of $Zerith1$.

**Observations Space** The observation space is composed of the following components: $p_t$ includes base linear velocity and PD gains; $e_t$ consists of push force, friction, mass parameters, height map, and the contact state of the foot end; and $o_t$ contains periodic signal input, desired velocity commands, joint position($q$), joint velocity($\dot{q}$), base angular velocity($\omega_{xyz}$), and base euler angles in the coordinate system ($\theta_{xyz}$).

**Action Space** The dimension of the action spaces $\mathcal{A}$ equals the number of actuators. The movement of each actuator is formulated as the bias between the target joint position $\theta_{\text{target}}$ and the nominal joint position $\theta_0$. The robot's target joint angle is defined as: $\theta_{\text{target}} = \theta_0 + a_t$.

### 3.2.2 DYNAMICS MODEL

The dynamics model predict the next time state based on history observation state. The backbone is an aggregation model $f_\psi$ that compresses the observation state $s_t^H$ into a hidden state sequence $h_t^H$.

The dynamics estimation model predicts $s_{\hat{t+1}}$ based on the hidden variable $h_t$. The dynamics model consists of these components:

$$
\begin{aligned}
\text{Aggregation Model: } & h_t^H = f_\psi(s_t^H) \\
\text{Dynamics Prediction Model: } & \hat{s}_{t+1} = p_\psi(h_t)
\end{aligned}
\tag{1}
$$

The aggregation model $f_\psi$ is implemented as a causally masked Transformer-XL, while $p_\psi$ is a linear layer. Transformer-XL introduces a recurrence mechanism that reuses the hidden states from the previous batch. This design overcomes the fixed-length limitation of traditional Transformer models, as highlighted in Lu et al. (2024), allowing the model to process longer sequences efficiently. By integrating immediate dynamic changes from environmental interactions with long-term dependencies in time series, the model achieves enhanced predictive accuracy.

### 3.2.3 PHYSICAL INTERACTION REGRESSION MODEL

We assume 1). the critic can access the full observation of the environment, 2). the hidden variable at time $t$ has learned the historical observation information before time $t$. We believe that the latent variable $h_t$ contains physical interaction information, and the regression model assists the actor network in learning this information. The regression model incorporates an aggregation model in Equation(2)., which encodes partial observation information $[o_t^H, a_{t-1}^H]$ into a hidden variable $z_t^H$.

$$
Aggregation\ Model\colon z_t^H = f_\psi(o_t^H, a_{t-1}^H)
\tag{2}
$$

$f_\psi$ is also implemented as Transformer-XL. Specifically, as referenced in Equation(4), physical interaction regression model employs a regression approach that utilizes the complete observation information provided by the critic network to guide the actor network in optimizing its latent variables. This process expands the observation space of the actor network, compensates for the limitations of partial observations, and enables the agent to better understand environmental dynamics and interaction relationships.

### 3.2.4 POLICY LEARNING

The actor network describes a Gaussian distribution based on the output mean and variance of the action, and then generates a specific action value by sampling from this distribution $a_t \sim \pi(a_t|o_t^H)$. The Critic network estimates the expected cumulative return $R_t$ under the current policy at state $s_t$: $v_\psi(R_t \mid s_t)$. The key to policy optimization lies in minimizing the error between the predicted value and the actual return $R_t$. By continuously optimizing this loss, the critic network is trained to more accurately evaluate state values. The key distinction from previous work lies in the introduction of time sequences and a world model for future prediction in our critic network not just actor network. This approach significantly enhances the critic's ability to evaluate the robot's state, thereby guiding decision-making with a more global perspective.

### 3.2.5 LOSS FUNCTION

Our loss function includes the dynamics model loss, the reconstruction loss for hidden variable regression, and the policy optimization loss. In each iteration, we first update the dynamics model and the PPO module, followed by optimizing the regression module.

**Dynamics Model Loss:** Our goal is to ensure that the dynamics estimation model can accurately predict future observation state. Inspired by the balanced cross-entropy loss used in (Robine et al., 2023), we also calculate the entropy and cross-entropy. We use the cross-entropy $\mathcal{L}_{ent2}$ of the dynamics prediction model to prevent the encoder from deviating from the dynamics model. Entropy $\mathcal{L}_{ent1}$ regularizes the latent states and prevents them from collapsing into a one-hot distribution. The dynamics predictor $\mathcal{L}_{NLL}$ is optimized via negative log-likelihood, providing rich learning signals for the latent states.

$$\mathcal{L}_{NLL} + \mathcal{L}_{ent1} + \mathcal{L}_{ent2} = \mathbb{E}\left[\sum_{t=1}^{T} -\ln P_\Phi(\hat{s}_{t+1}|h_t) + \alpha_1 H(P_\Phi(h_t|s_t)) + \alpha_2 H(P_\Phi(s_{t+1}), P_\Phi(\hat{s}_{t+1}|h_t))\right] \tag{3}$$

Hyperparameters $\alpha_1, \alpha_2$ are the relative weights of the terms.

**Reconstruction Loss:** This loss corresponds to the regression model described in Section 3.2.3, where the latent variable $h_t$ generated by the critic network supervises the learning of the latent variable $z_t$ produced by the actor network. The mean squared error (MSE) loss we adopt for this purpose is as follows:

$$\mathcal{L}_{reconstruct} = MSE(z_t, h_t) \tag{4}$$

**Policy Optimization Loss:** We use the Proximal Policy Optimization (PPO) algorithm to optimize the policy. The loss function primarily consists of a policy loss and a value function loss, with an entropy term added to encourage policy diversity. The objective of policy optimization is to update the policy using a surrogate loss, which can be expressed as follows:

$$\mathcal{L}^{clip}(\theta) = E_t\left[\min\left(r_t(\theta)\hat{A}_t, \text{clip}\left(r_t(\theta), 1-\epsilon, 1+\epsilon\right)\hat{A}_t\right)\right] \tag{5}$$

$r_t(\theta) = \frac{\pi_\theta(a_t|o_t^H)}{\pi_{\theta_{old}}(a_t|o_t^H)}$ represents the ratio between the new and old policy, $\hat{A}_t$ is the advantage function that quantifies how much better the current action is compared to the average, and $\epsilon$ is the clipping parameter that controls the degree of policy updates. The following loss function is utilized to optimize the value function:

$$\mathcal{L}_{value} = (V(s_t) - R_t)^2 \tag{6}$$

The overall training loss is defined as

$$\mathcal{L} = \mathcal{L}^{clip}(\theta) + \mathcal{L}_{value} + \mathcal{L}_{NLL} + \mathcal{L}_{ent1} + \mathcal{L}_{ent2} + \mathcal{L}_{reconstruct} \tag{7}$$

## 4 EXPERIMENTS

### 4.1 EXPERIMENT SETTING

**Benchmark Comparision.** For a comparative evaluation, the experiments we performed are as follows:

- **Oracle**: Train the policy with a history of full privileged observations.
- **Baseline**: MLP network optimized using the PPO algorithm.
- **LSTM**: Adopt LSTM as network backbone Siekmann et al. (2020)
- **Bert**: We implement the policy according to the Humanplus algorithm Fu et al. (2024), Compared to Transformer-XL, the like-Bert structure lacks memory information and only focuses on the current time window.
- **Ours w/o estimator**: The proposed method without dynamics estimation module.
- **Ours w/o wm** : The proposed method without dynamics estimation module and latent variable reconstruction.
- **Ours w/o regression**: The proposed method without latent variable reconstruction.

**Setups in Simulations.** We conduct simulation experiments on the Isaac Gym platform, training 4096 agents in parallel using domain randomization. We test performance by comparing the convergence curves of rewards, the convergence curves of terrain levels, and the velocity tracking under various terrains. Details about the ablation experiments can be found in Appendix A.3. The training is conducted on an NVIDIA V100 GPU with 40 GB of memory. Details on domain randomization and reward design can be found in Appendix A.2.

**Setups in Real-world.** In this study, we employed a lightweight small-scale robot named $Zerith1$ as a testing platform. The detailed information of the robot is shown in Fig. 3.

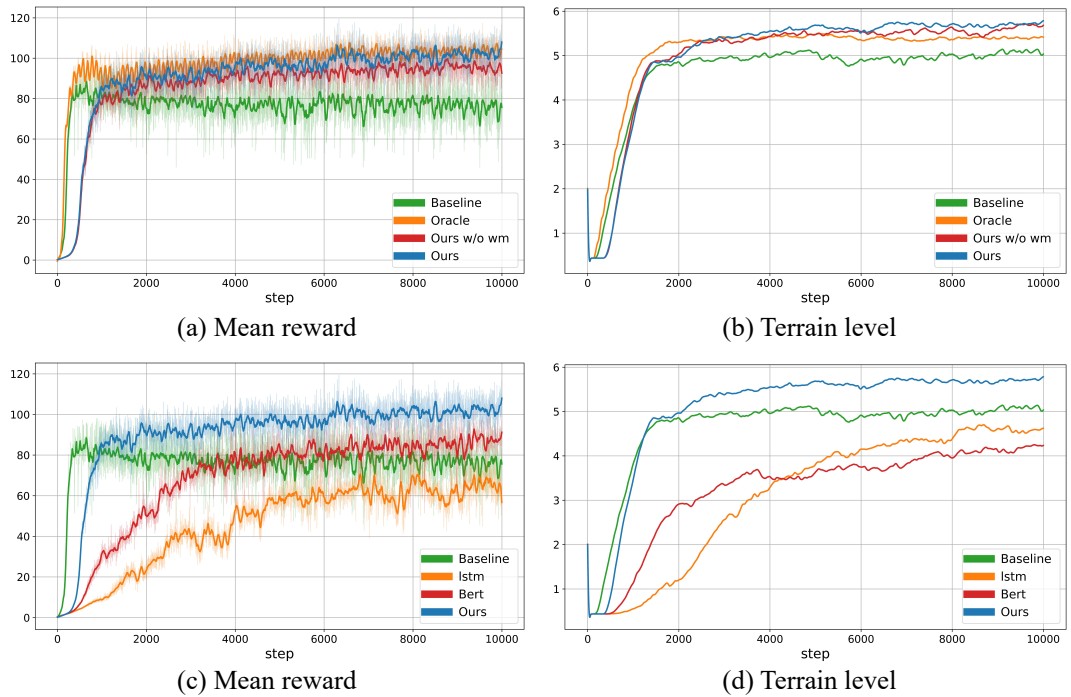

Figure 4: Comparison of different Method. (a) and (b) are compared with the baseline, oracle, and ablation experiments in terms of terrain levels and average rewards to demonstrate model performance, while (c) and (d) are compared with other methods to showcase the superiority of our model. We adopt curriculum learning Bengio et al. (2009) for training. Terrain level refers to the difficulty level of the terrain.

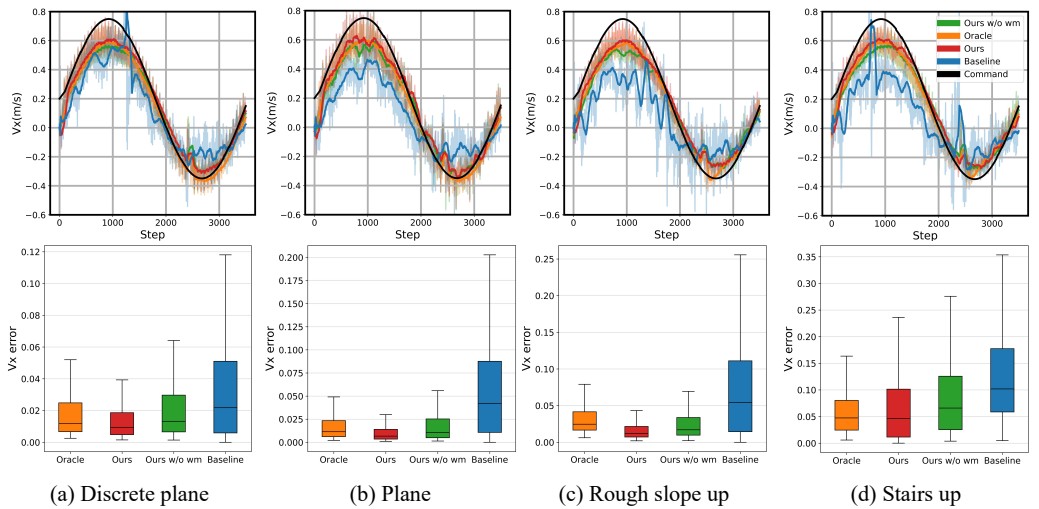

Figure 5: Vehicle tracking comparision. We provided the robot with a sinusoidal velocity command and tested the average velocity of 100 robots on different terrains. The Vx error is calculated using the following formula: $\text{Vx\_error} = \frac{1}{N} \sum_{i=1}^{N} \left( V_{x,\text{command}}(t) - V_{x,i}(t) \right)^2$
.

**Terrain Passability Experiment:** We first tested the upper limit and robustness of our method across various complex terrains. As shown in Fig.4, our method significantly outperforms the baseline in handling complex terrains compared to a simple MLP structure. Additionally, our method surpasses even the "oracle" method, which has access to privileged information, in terms of the final terrain difficulty. This demonstrates that the transformer architecture effectively utilizes the

robot's historical information to enhance decision-making. Our method also outperforms the ablated version, highlighting the importance of the world model in understanding dynamic interactions, allowing the robot to navigate complex terrains more efficiently and stably. The comparison with other methods further demonstrates that our approach is more robust and adaptable to different challenging terrains.

**Command Tracking Experiment:** We also quantitatively evaluated the ability of our method to track desired velocities in complex terrains. As shown in Fig.5, (a), (b), (c), and (d) compare the velocity tracking performance of different methods across various terrains. The top four plots show the actual velocity feedback curves as the robot tracks a continuously changing sine-wave desired velocity, while the bottom four plots present boxplots of the tracking errors in the x-direction for different methods. Our method demonstrates superior tracking performance across various terrains. In terms of both the upper bound of error and the median, our method significantly outperforms other methods. Even though the Oracle has access to foot elevation maps, HuWo outperforms Oracle on discrete, plane, and slope terrains. On stair terrains, which rely on foot elevation data, our method performs close to Oracle, indicating that the environmental estimation of our world model is already very close to the actual elevation map.

**Latent Layer Analysis:** As the robot transitions through a plane-slope-plane terrain environment, we visualized the outputs of 6 selected neurons from the 128-dimensional hidden layer. As shown in Fig.6, the changes in hidden layer responses during terrain transitions highlight the robot's ability to adapt to varying terrains. These responses reflect the network's capability to recognize and respond to terrain changes, enabling real-time adjustments to the robot's control strategy.

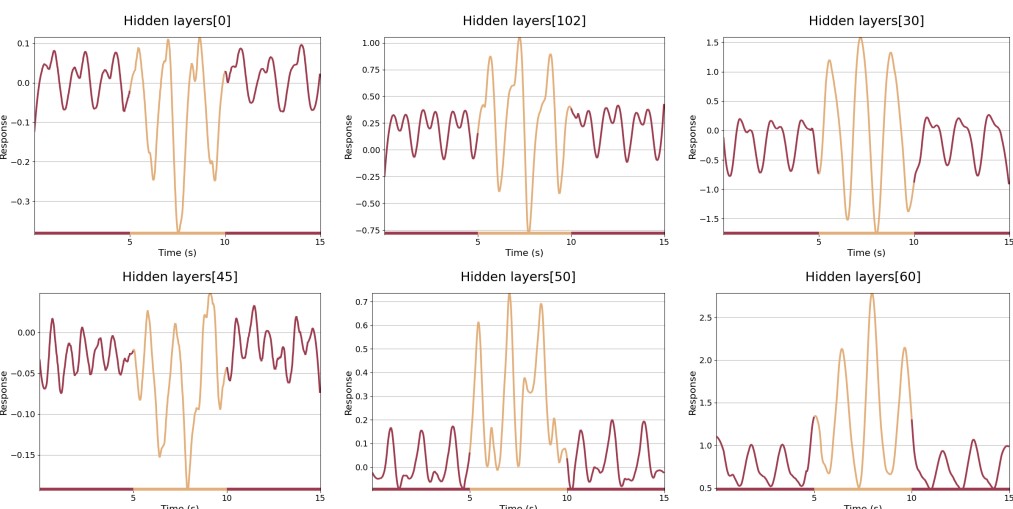

Figure 6: Hidden layers visualization. The figure shows the changes in part of the hidden layer responses as the robot moves from flat ground to slope up and back to flat ground. The red line corresponds to the time when the robot is walking on flat ground, while the yellow line corresponds to the time when it is slope up.

## 4.2 REAL-WORLD RESULTS

In the physical experiments, we primarily conduct qualitative analyses to verify whether the robot possesses self-awareness and environmental perception. We first tested the robot's traversal capability across various complex terrains. As shown in Fig.1, 8 and 7, the robot successfully navigated challenging environments such as grassy fields, rocky ground, sandy surfaces, and terrains representing different hardness and friction coefficients (a soft rubber track, a relatively smooth wooden bridge, and a hard asphalt road), as well as five-level stairs and artificially constructed complex terrains. This demonstrates that our approach enables the robot to utilize only proprioception and historical data to accurately assess different environments and make appropriate decisions.

Fig.7 particularly highlights the robot's correct estimation of the elevation map around its feet. During normal walking, the robot does not maintain a high-stepping gait to conserve energy, which

aligns with human walking intuition. However, when its feet encounter an obstacle, the robot quickly raises its foot, allowing it to overcome a 10 cm stair. This proves that our method can be effectively transferred to a real robot, allowing it to retain accurate self-awareness and environmental perception.

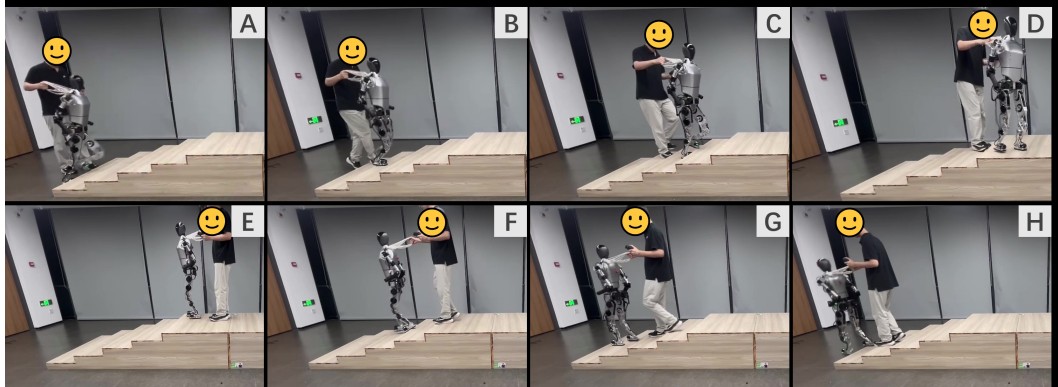

Figure 7: Stairs terrain: 30cm width, 10cm height, 5 steps in total. The tester only held the rope but did not apply any force to it

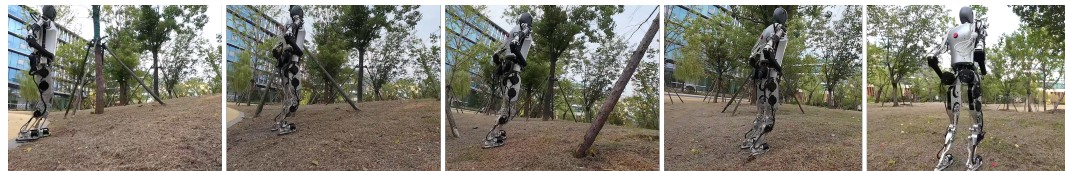

Figure 8: Humanoid traversing slope outdoors

In addition to understanding terrain, environmental perception also involves recognizing and predicting interactions with the environment. Therefore, we intentionally introduced disturbances to the robot to observe whether it could respond quickly. As shown in Fig.9, when the tester applied a significant force to the robot's foot, disrupting its static posture, the robot quickly adjusted its body position to regain balance and return to a stationary state. This demonstrates that our method also exhibits strong robustness when applied to real-world robots.

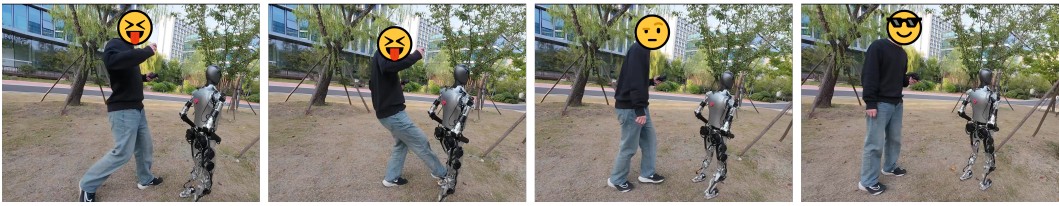

Figure 9: Disturbance experiments

## 5 CONCLUSION

In this work, we propose Interactive World Models for Humanoid Robots (HuWo), a novel framework designed to tackle the challenges of humanoid robot locomotion in complex environments. Our framework leverages the world model concept within an asymmetric actor-critic structure, where the hidden layers of Transformer-XL enable end-to-end implicit modeling of dynamic interactions between the robot and its environment. This approach facilitates robust decision-making by expanding the observation space through historical sequences and dynamic predictions. The effectiveness of our method is demonstrated through experiments showing robust walking performance in diverse real-world environments, including zero-shot sim-to-real transfer. These results highlight our framework's ability to model environmental dynamics and adapt to changing conditions. Future work will focus on extending this framework to achieve full-body coordination, including free arm movement, for more versatile and natural locomotion.

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

# A  APPENDIX

## A.1  HYPERPARAMETERS OF HULO

Out detailed Netowrk hyperparameters are shown in Table 1. We a single NVIDIA V100 GPU, we simultaneously trained 4096 domain-randomized $Zerith1$ robot environments in Isaac Gym. During training, we employed PD position controllers in 1000 HZ for each joint, with the Policy running at a frequency of 100 Hz.

Table 1: Hyperparameters of Hulo

| Parameter | Value |
|---|---|
| Number of Environments | 4096 |
| Context window | 8 |
| Memory window | 8 |
| Batch size | $4096 \times 24$ |
| Discount Factor | 0.99 |
| GAE discount factor | 0.95 |
| Entropy Coefficient | 0.00001 |
| PPO lr | 0.0001 |
| $\alpha_1$ | 5.0 |
| $\alpha_2$ | 0.01 |
| Transformer blocks | 4 |
| Embedding dimension | 128 |
| Multi-head attention heads | 4 |
| Reconstruction module lr | $1 \times 10^{-6}$ |
| Dynamic estimator module lr | $1 \times 10^{-6}$ |

## A.2  TRAINING DETAILS

We used the reward function as shown in Table 2, where the Task reward guides the robot to track the desired speed and complete motions on various terrains and alive reward mitigates the exploration burden in early period. Besides, we design comprehensive reward about feet( Siekmann et al. (2021), Margolis & Agrawal (2023)) to guide locomotion through tough terrain and prevent weird posture. Through extensive training trials, we optimized our reward weight settings to ensure that the robot moves in a relatively ideal manner. The domain randomizations and terrain setting details are in Table 3 and 4

## A.3  ABLATION EXPERIMENTS

In Fig. 10, In the self-ablation experiment, we compared our method with the ablated versions and found that the latent variable regression part and the future information prediction part influence each other. Having both components leads to better performance, which is understandable. The key to our approach lies in introducing time series through the critic and leveraging the world model for future predictions. This method enhances the evaluation capability of the critic network, guiding better decision-making abilities.

As shown in Fig.11 and Fig.12, we experimented with varying history length and hidden layer dimensions to verify whether our parameters achieve optimal locomotion performance and robustness. The time window determines the context range the model can observe when handling sequential tasks. A larger window helps capture long-range dependencies but increases computational costs. The model's performance is similar when the window length is 16 and 8, and significantly better than other window lengths. The hidden layer size determines the model's representation capacity, and increasing the number of hidden layers helps improve the network's fitting ability. The perfor-

| Term | Equation | Weight |
|------|----------|--------|
| **Task Reward** | | |
| alive | $1$ | 0.5 |
| xy velocity tracking | $\exp\{-|\mathbf{v}_{xy} - \mathbf{v}_{xy}^{\mathrm{cmd}}|^2 * 5\}$ | 1.5 |
| yaw velocity tracking | $\exp\{-(\boldsymbol{\omega}_z - \omega_z^{\mathrm{cmd}})^2 * 5\}$ | 1.0 |
| **Feet Guidance** | | |
| swing phase tracking (force) | $\sum_{\mathrm{foot}}[1 - C_{\mathrm{foot}}^{\mathrm{cmd}}(\boldsymbol{\theta}^{\mathrm{cmd}}, t)]\exp\{-|\mathbf{f}^{\mathrm{foot}}|^2/100\}$ | 5.0 |
| stance phase tracking (velocity) | $\sum_{\mathrm{foot}}[C_{\mathrm{foot}}^{\mathrm{cmd}}(\boldsymbol{\theta}^{\mathrm{cmd}}, t)]\exp\{-|\mathbf{v}_{xy}^{\mathrm{foot}}|^2/5\}$ | 10.0 |
| raibert heuristic footswing tracking | $(\mathbf{p}_{x,y,\mathrm{foot}}^{f} - \mathbf{p}_{x,y,\mathrm{foot}}^{f,\mathrm{cmd}}(\boldsymbol{s}_y^{\mathrm{cmd}}))^2$ | $-30.0$ |
| footswing height tracking | $\sum_{\mathrm{foot}}(\boldsymbol{h}_{z,\mathrm{foot}}^{f} - \boldsymbol{h}_z^{f,\mathrm{cmd}})^2 C_{\mathrm{foot}}^{\mathrm{cmd}}(\boldsymbol{\theta}^{\mathrm{cmd}}, t)$ | $-10.0$ |
| **Regularization Reward** | | |
| body height | $\exp\{-(\boldsymbol{h_z} - \boldsymbol{h}_z^{\mathrm{cmd}})^2 * 1000\}$ | $-0.2$ |
| z velocity | $\mathbf{v}_z^2$ | -0.02 |
| foot slip | $|\mathbf{v}_{xy}^{\mathrm{foot}}|^2$ | -0.04 |
| hip position | $\exp\{-\sum_{i=1}^{2} q_{roll,yaw}^2 * 100\}$ | 0.4 |
| feet orientation | $\exp\{-\sum_{i=1}^{2} |\theta_{roll,pitch}^{\mathrm{foot}}| * 10\}$ | 0.4 |
| feet stumble | $\mathbb{1}(\max_i(\sqrt{F_{x_i}^2 + F_{y_i}^2} > 4|F_{z_i}|))$ | -1 |
| orientation | $\exp\{-|g_{xy}|^2 * 10\}$ | 1.5 |
| thigh/calf collision | $1_{\mathrm{collision}}$ | $-5.0$ |
| joint limit violation | $1_{q_i > q_{max} || q_i < q_{min}}$ | $-10.0$ |
| joint torques | $|\boldsymbol{\tau}|^2$ | -1e-5 |
| joint velocities | $|\dot{\mathbf{q}}|^2$ | -1e-3 |
| joint accelerations | $|\ddot{\mathbf{q}}|^2$ | -2.5e-7 |
| action rate | $|\mathbf{a}_t|$ | -5e-5 |
| action smoothing | $|\mathbf{a}_{t-1} - \mathbf{a}_t|^2$ | -0.01 |
| action smoothing, 2nd order | $|\mathbf{a}_{t-2} - 2\mathbf{a}_{t-1} + \mathbf{a}_t|^2$ | -0.01 |

Table 2: Reward Function

Table 3: Domain Randomizations and their Respective Range

| Parameters | Range [Min, Max] | Unit |
|------------|------------------|------|
| Ground Friction | [0.1, 1.5] | - |
| Ground Restitution | [0.0, 0.25] | - |
| Body Mass | [-2, 5] | Kg |
| Body Com | [-0.07, 0.1] | Kg |
| Link Mass | [0.8, 1.4] × nominal value | Kg |
| Joint $K_p$ | [0.85, 1.15] × 20 | - |
| Joint $K_d$ | [0.85, 1.15] × 0.5 | - |
| System Delay | [0, 40] | ms |
| External Force | interval = 5s $vel_{xy} = 0.4$ | - |

mance is similar when the number of hidden units is 256 and 128, with the model showing slightly better exploration ability in complex terrains when the hidden layer size is 128.

## A.4 PHYSICAL INTERACTION WORLD MODEL VS DENOISING WORLD MODEL

Using the $Zerith1$ model, we compared the terrain level progression curves of our method with those of the Denoising World Model Gu et al. (2024) approach. As shown in Fig.13, our method achieves significantly faster and higher terrain level progression, highlighting its superior capability to handle more complex terrains. These findings demonstrate that our approach, powered by the $Zerith1$ robotic model, outperforms the Denoising World Model in adapting to challenging environments.

Table 4: Terrain Setting Range

| Parameters | Range [Min, Max] | Proportion |
|---|---|---|
| Stair up | [5cm, 12cm] | 0.5 |
| Stair down | [5cm, 12cm] | 0.5 |
| Slop up | [0, 0.2] | 2.5 |
| Slop down | [0, 0.2] | 1 |
| Plane | - | 0.5 |

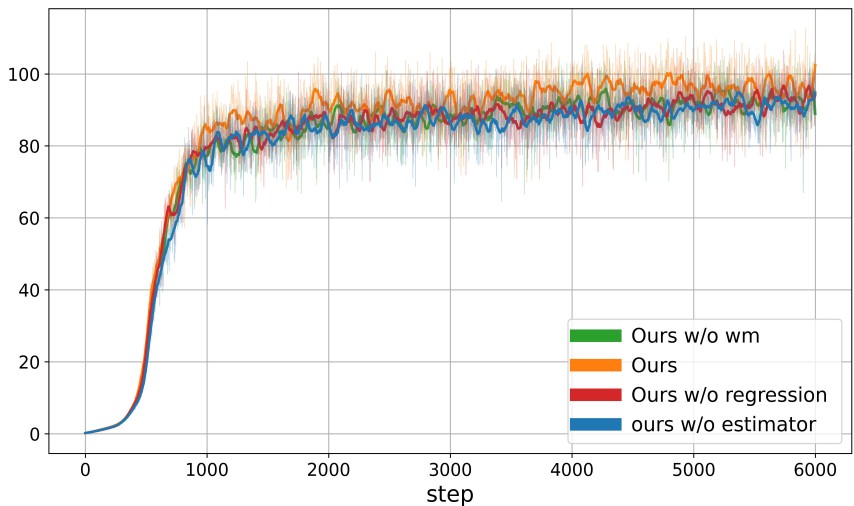

Figure 10: Self-ablation experiments

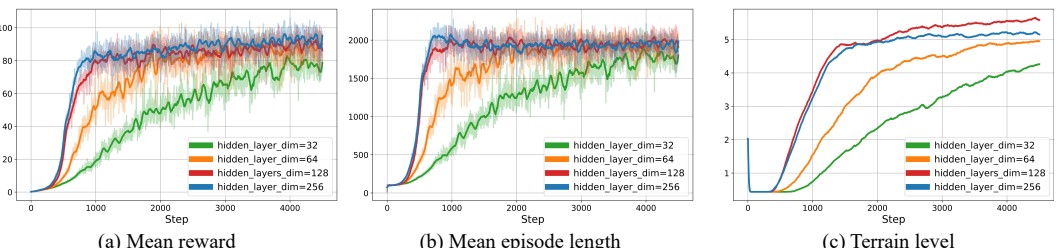

(a) Mean reward  (b) Mean episode length  (c) Terrain level

Figure 11: The effect of the dimension of hidden layer

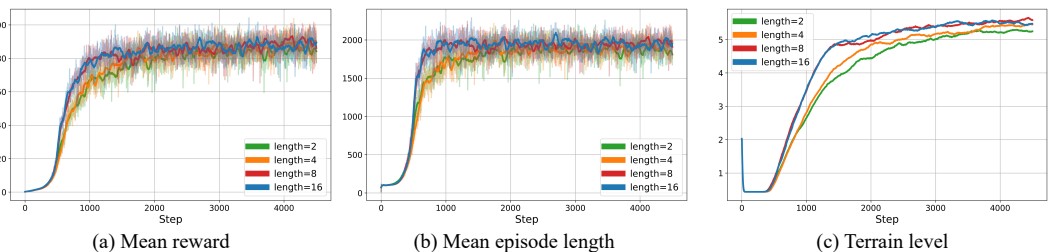

(a) Mean reward  (b) Mean episode length  (c) Terrain level

Figure 12: The effect of time window length

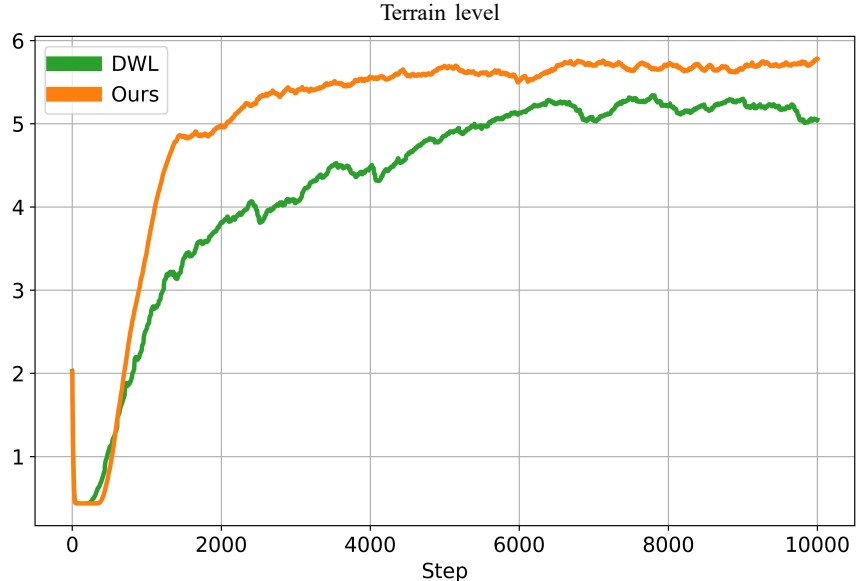

Figure 13: Ours vs Denoising World Model

