# OpenReview forum: "HuWo：Building Physical Interaction World Models for Humanoid Robot Locomotion"
_ICLR.cc/2025/Conference — Submitted to ICLR 2025_

### Official Review · Reviewer_QaRB · 2024-10-24

**Soundness:** 3
**Presentation:** 3
**Contribution:** 3
**Rating:** 6
**Confidence:** 4

**Summary:**

This paper presents an approach to train a reinforcement learning-based controller for a humanoid robot. The key components are (1) Transformers for both actor and critic, which allows for efficient learning from the history of observation (2) in addition to learning the value function, the critic is also tasked with learning the system dynamics from the learned hidden state. (3) The actor is also tasked to match the hidden state of the critic.

**Strengths:**

The system is straightforward to understand, and the presentation is good.

The proposed method can learn robust control policies for a humanoid robot.

Analysis of the hidden state.

**Weaknesses:**

The paper should be revised for some typos, e.g., in the paragraph Terrain Passability Experiment, there is a missing reference to a figure.

I am not sure about some of the points, which I will list below.

**Questions:**

From Fig 4, it does not look like learning a world model helps much. The yellow and green curves converge to similar values and in almost the same number of steps.

What is the difference between a world model and a dynamics estimator? It would be nice to clearly point out which part of the system is the world model and which part is the dynamics estimator in Fig. 2.

The ablation on the dynamics estimation module is missing. There is also no ablation on the latent variable regression. These ablations are mentioned in the text but are not in the figures.

---

> ### Author Response · Authors · 2024-11-24
> **Response to Reviewer QaRB**
>
> Thank you for your review and comments. Below are our responses to the listed weaknesses and questions:
> # Weaknesses:
> > The paper should be revised for some typos, e.g., in the paragraph Terrain Passability Experiment, there is a missing reference to a figure.
>
> We appreciate the reviewer for pointing out the typographical issues in the manuscript. Specifically, we have corrected the missing reference to the figure in the "Terrain Passability Experiment" section.
> # Questions:
> > Q1: From Fig 4, it does not look like learning a world model helps much. The yellow and green curves converge to similar values and in almost the same number of steps.
>
> Due to possible color differentiation issues, the distinction between our two curves may not have been clear; we have updated $\color{red}{Fig. 4 }$ for better clarity. From the updated figure, the blue curve (our method) outperforms the red curve (without the world model) in several aspects. In the "Mean reward" plot, the blue curve shows greater stability and slightly higher rewards, particularly after 4000 steps, indicating that the world model reduces training instability and improves final performance. In the "Terrain level" plot, the blue curve surpasses the red curve after 2000 steps, adapting to complex terrains faster and achieving higher final levels. These results highlight the advantages of incorporating the world model in terms of training efficiency, stability, and adaptability.
> > Q2: What is the difference between a world model and a dynamics estimator? It would be nice to clearly point out which part of the system is the world model and which part is the dynamics estimator in Fig. 2.
>
> Thank you for your feedback. The description of the world model, including its two key components—the dynamics estimation module and the physical interaction regression module—was mentioned in $\color{red}{line\ 239 }$ of the main manuscript. Below is a brief description of their roles: \
> $\bullet$ Dynamics Estimation Module: \
> This module predicts the system's next state based on the observed state, helping the agent understand how its actions influence future states. It is the core component of the world model, responsible for capturing the fundamental dynamics of the environment. \
> $\bullet$ Physical Interaction Regression Module: \
> This module leverages the complete observation information provided by the critic network to guide the actor network in optimizing its latent variables. This process expands the observation space of the actor network, compensates for the limitations of partial observations, and helps the agent better understand environmental dynamics and interaction relationships. \
> The details can be found in $\color{red}{Section\ 3.2.2. 3.2.3 }$. In addition, Thank you for suggesting an update to Fig. 2. After careful consideration, we decided not to modify the figure to preserve its readability and simplicity. Adding detailed information about the dynamics estimation and physical interaction regression modules directly in the figure could make it overly complex and harder to interpret.
> > Q3: The ablation on the dynamics estimation module is missing. There is also no ablation on the latent variable regression. These ablations are mentioned in the text but are not in the figures.
>
> We apologize for the lack of explicit references in the manuscript, which may have made it difficult to locate our ablation experiments. These experiments, including the ablation of the dynamics estimation module, latent variable dim, and time window length, are detailed in $\color{red}{Appendix\ A.3}$. We have addressed this oversight in the revised manuscript by ensuring clear references in the main text.

---

> ### Comment · Area_Chair_UsDo · 2024-11-25
>
> Dear Reviewer,
>
> Please provide feedback to the authors before the end of the discussion period, and in case of additional concerns, give them a chance to respond.
>
> Timeline: As a reminder, the review timeline is as follows:
>
> November 26: Last day for reviewers to ask questions to authors.
>
> November 27: Last day for authors to respond to reviewers.

---

### Official Review · Reviewer_AAFX · 2024-11-02

**Soundness:** 3
**Presentation:** 2
**Contribution:** 3
**Rating:** 5
**Confidence:** 4

**Summary:**

This paper introduces HuWo, an end-to-end reinforcement learning framework that leverages a physical interaction model using Transformer-XL to enhance humanoid robot locomotion in complex environments. The approach implicitly models dynamic interactions between the robot and its environment across temporal sequences. HuWo demonstrates robustness and flexibility in challenging terrains like slopes, stairs, and discontinuous surfaces, outperforming baseline models in traversability and command tracking.

**Strengths:**

Originality: The authors cleverly integrate the concepts of predicting states with a dynamics model within an RL training framework. The idea makes perfect sense to model the complex dynamics between the humanoid and the environment.

Quality: The gait appears robust in most parts of the video, and I commend the authors for a thorough evaluation across diverse terrains. The baseline comparisons in the simulation are also convincing.

Clarity: The paper is clear and easy to understand.

Significance: This method effectively combines the strengths of both world-model-based approaches and powerful RL frameworks with domain randomization. The learned dynamics model is well-suited for capturing the complex interactions between the humanoid and its environment.

**Weaknesses:**

1. The presentation could be improved with more attention to figure and text quality. For instance, the text in many figures is too small and difficult to read without close inspection, and there is a missing figure reference on page 8. Additionally, the formatting and choice of symbols in mathematical equations could be refined for better readability. Some claims in the paper are also questionable, such as: “However, for humanoid robots, these methods can only handle relatively simple environments and have not yet fully addressed dynamic control issues in complex terrains.” Prior research has shown that RL can handle challenging terrains for humanoid robots [1,2,3].

1. The evaluation would be more robust with additional quantitative results from real-world experiments. The sim-to-real gap can be significant, and while the authors show the policy performs in both simulation and real settings qualitatively, simulation results alone may not fully demonstrate real-world effectiveness without quantitative results, as performance could differ substantially.

1. Finally, the gait in the supplementary video is not particularly impressive, as humanoid walking videos have become common. The stride length is short, and the speed seems slow relative to the robot’s scale. In the stair-climbing experiment at the end, the walking also appears less robust.

[1] Gu et al., *Advancing Humanoid Locomotion: Mastering Challenging Terrains with Denoising World Model Learning*, RSS 2024.

[2] Liao et al., *Berkeley Humanoid: A Research Platform for Learning-based Control*, ArXiv 2024.

[3] Zhuang et al., *Humanoid Parkour Learning*, CoRL 2024.

**Questions:**

1. Could you motivate more why the dynamics model helps learn humanoid locomotion? Since as I mentioned above, RL approaches seem to already work well enough with recent advances. Some reasons I can image include data efficiency or better generalization capabilities by incorporating a world model. But if that's the case, more evidence needs to be shown to support these claims.

1. Could you show more evidence that the proposed method is superior to baseline methods in the real world? For example, what is the velocity tracking performance in the real world? Have you considered *Gu et al.* as a baseline?

1. In Figure 4, what does terrain level mean? Is there a more convincing metric than mean rewards?

**Details Of Ethics Concerns:**

It’s a bit unusual that the colon in the title resembles a Chinese-style colon, which may introduce some bias if the reviewers happen to speak Chinese. Additionally, I noticed that the human operator’s face is not covered in the video. This might raise concerns regarding the double-blind review guidelines.

---

> ### Author Response · Authors · 2024-11-25
> **Response to Reviewer AAFX**
>
> Thank you for your review and comments. To respond/comment on your listed weaknesses:
> # Weaknesses:
> > W1: The presentation could be improved with more attention to figure and text quality. For instance, the text in many figures is too small and difficult to read without close inspection, and there is a missing figure reference on page 8. Additionally, the formatting and choice of symbols in mathematical equations could be refined for better readability. Some claims in the paper are also questionable, such as: “However, for humanoid robots, these methods can only handle relatively simple environments and have not yet fully addressed dynamic control issues in complex terrains.” Prior research has shown that RL can handle challenging terrains for humanoid robots [1,2,3].
>
> $\bullet$ We appreciate the reviewer’s feedback regarding figure and text quality. We have reviewed and updated all figures to ensure the text is legible. Specifically, the font size of figure labels and captions has been increased, and we have ensured proper alignment and clarity. Additionally, the missing figure reference on page 8 has been corrected. \
> $\bullet$ Currently, learning-based humanoid robot locomotion methods still face significant challenges in terms of task success rates and the complexity of tasks they can handle. Even in industrial applications, achieving long-term stable locomotion for humanoid robots on complex terrains remains a major challenge. Regarding the works mentioned by the reviewer as being highly effective, we provide the following analysis: \
> [3] relies on external sensors and depth cameras to accomplish complex tasks, which introduces dependency on additional hardware. \
> [2] focus of this study is on designing minimalistic reinforcement learning controllers to validate the sufficiency of hardware designs based on learning-based control. However, the terrains involved in their experiments are not particularly challenging. \
> [1] proposes Denoising World Model Learning, while our approach employs a fundamentally different world model. In the revised manuscript, we have added comparative experiments in simulation to evaluate the terrain level progression curves of both methods. $\color{red}{Fig. 13}$ demonstrates that our method achieves a faster progression in terrain complexity.
>
> > W2: The evaluation would be more robust with additional quantitative results from real-world experiments. The sim-to-real gap can be significant, and while the authors show the policy performs in both simulation and real settings qualitatively, simulation results alone may not fully demonstrate real-world effectiveness without quantitative results, as performance could differ substantially.
>
> Thank you for your feedback and for emphasizing the importance of quantitative results from real-world experiments for evaluation. We fully understand that real-world assessments can provide more robust validation of the proposed method.
>
> However, due to the safety concerns and high risks involved in testing humanoid robots in unstructured environments, conducting large-scale real-world experiments is not feasible within the scope of this study. Instead, we have focused on demonstrating the zero-shot sim-to-real transfer capability of our method through qualitative real-world experiments and robust testing in simulation.
>
> Although we have not provided quantitative real-world results, we have ensured that our simulation environment closely mirrors real-world scenarios. This includes domain randomization, curriculum learning, and leveraging dynamic prediction to reduce the sim-to-real gap. These techniques collectively improve the generalization and robustness of the proposed method across diverse and unpredictable environments.
>
> > W3: Finally, the gait in the supplementary video is not particularly impressive, as humanoid walking videos have become common. The stride length is short, and the speed seems slow relative to the robot’s scale. In the stair-climbing experiment at the end, the walking also appears less robust.
>
> Thank you for your feedback on the gait and stair-climbing experiments in our supplementary video. While the robot’s stride length and speed may not appear remarkable, our primary focus is to demonstrate the proposed framework's adaptability and robustness in challenging real-world scenarios, rather than optimizing gait parameters such as speed or stride length. \
> The stair-climbing experiment specifically highlights the robot's ability to navigate uneven and unpredictable terrain, showcasing the strength of our method in handling complex dynamics and partial observability—key factors for reliable locomotion in unstructured environments. \
> We appreciate your observations and will explore further optimization of gait parameters in future work.

---

> > ### Author Response · Authors · 2024-11-25
> > **Response to Reviewer AAFX**
> >
> > # Questions:
> > > Could you motivate more why the dynamics model helps learn humanoid locomotion? Since as I mentioned above, RL approaches seem to already work well enough with recent advances. Some reasons I can image include data efficiency or better generalization capabilities by incorporating a world model. But if that's the case, more evidence needs to be shown to support these claims.
> >
> > Our proposed Physical Interaction World Models method includes a dynamics model and a physical interaction regression model. Motivation for the Interaction World Model in Humanoid Locomotion:
> > $\bullet$ Reducing the Sim-to-Real Gap: The dynamics model predicts the next observation based on historical data, and a loss is computed by comparing the predicted and actual observations. This process helps the model align better with real-world dynamics.  $\color{red}{Section\ 3.2.2}$
> >
> > $\bullet$ Learning Physical Interaction Information: Unlike prior methods [4][5] that focus on predicting static environmental or robot dynamics, our approach enables the actor network to learn temporally correlated interaction information between the robot and its environment via the physical interaction regression model. Leveraging a Transformer-based architecture, the model effectively captures both dynamic and long-term dependencies. $\color{red}{Section\ 3.2.3}$
> >
> > > Could you show more evidence that the proposed method is superior to baseline methods in the real world? For example, what is the velocity tracking performance in the real world? Have you considered Gu et al. as a baseline?
> >
> > As shown in $\color{red}{Fig. 5}$, the baseline method demonstrates poor velocity tracking performance in simulation, indicating its limited adaptability to dynamic environments. Furthermore, given its suboptimal performance in simulation, directly deploying it to a real robot is unlikely to succeed and could pose significant safety risks in unstructured environments.
> > In our response to W1, we have discussed comparative experiments with the method proposed in [1] within a simulation setting.
> > If time and hardware resources permit, we will supplement our work with quantitative experiments in real-world scenarios to further validate the method's performance on physical robots.
> > > In Figure 4, what does terrain level mean? Is there a more convincing metric than mean rewards?
> >
> > Thank you for pointing out the need for further clarification of the metrics. In Figure 4, terrain level represents the complexity of the terrain successfully navigated by the agent, reflecting its adaptability and generalization capabilities.
> > In addition to mean rewards, we also provide mean episode length in Appendix A3, which measures the agent's average survival time before failure, reflecting the robustness of the policy. Furthermore, the velocity tracking performance and tracking error metrics in Figure 5 evaluate the agent's ability to follow target commands under various terrain conditions.
> > We encourage the reviewer to refer to $\color{red}{Fig.4}$、$\color{red}{Fig.5}$、$\color{red}{Appendix\ A.3}$ for further details.
> >
> > # Reference
> > [1] Gu et al., Advancing Humanoid Locomotion: Mastering Challenging Terrains with Denoising World Model Learning, RSS 2024.
> >
> > [2] Liao et al., Berkeley Humanoid: A Research Platform for Learning-based Control, ArXiv 2024.
> >
> > [3] Zhuang et al., Humanoid Parkour Learning, CoRL 2024.
> >
> > [4] I Made Aswin Nahrendra et al., DreamWaQ: Learning Robust Quadrupedal Locomotion With Implicit Terrain Imagination via Deep Reinforcement Learning. ICRA 2023.
> >
> > [5] Cui et al., Adapting Humanoid Locomotion over Challenging Terrain via Two-Phase Training. CORL 2024.

---

### Official Review · Reviewer_Amo3 · 2024-11-04

**Soundness:** 2
**Presentation:** 2
**Contribution:** 2
**Rating:** 6
**Confidence:** 3

**Summary:**

The paper presents an asymmetric actor-critic framework to train a policy for blind bipedal locomotion, based on the transformer-XL architecture.  In particular, the critic is trained (with privileged information) as a value function approximator and also as a world model that predicts the next state observation, based on the hidden variables $h_t$. Such privileged hidden variables are then used to supervise the hidden variables $z_t$ of the actor, which are generated from observations with corrupted information and are also used to obtain the next action. The method is trained in simulation and demonstrated on hardware, on a set of challenging tasks.

**Strengths:**

- Blind bipedal locomotion is a challenging problem, which the paper successfully addresses, bridging the sim-to-real gap and demonstrating a good performance of the trained policy on hardware.

- The proposed method outperforms the evaluated baselines, in general, in terms of overall reward and velocity tracking performance.

**Weaknesses:**

- The paper is good as a systems paper. However, my main concern is that the conference might not be a good fit for the paper. The scope of the paper is very specific to one domain and while the proposed method is potentially useful for other applications, showcasing its usefulness in different domains will require further experiments on a broader set of environments and tasks, which is outside the scope of the rebuttal period.

- The paper would benefit from qualifying,  justifying, or explaining in more detail statements like:

  - L 223: “After establishing an understanding of the interaction process with the environment, the robot should learn and comprehend interaction information during dynamic prediction.”  What do you mean by “understanding of the interaction process with the environment”? What do you mean by “comprehending interaction information during dynamic prediction?” How can you evaluate that?

  - L 230. “However, partial observations cannot withstand the challenges posed by complex environments”. I guess, the sentence is trying to make a point about the fact that policies that receive only partial information do not perform well in complex environments and that a history of partial observations helps to improve the performance of the policy.  However, the sentence is not clear by itself.

  - L318: “Learn richer and more precise representations in the latent space”.  How can a latent representation be more precise?



- The paper has several typos, many of them related to missing spaces before citations. To mention a few:
  - L062: Missing word - complex non-linear* dynamics.
  - L107: Missing spaces - learning_Escontrela
  - L112: Typo - to solve* this problem.
  - L113: Missing space -  (2023)_that utilizes
  - L116: Missing space - such as gait_Margolis
  - L419: Broken reference Fig ??.

**Questions:**

- What is the rationale behind the order of the updates? After optimizing the dynamics models, why is the policy optimized first and not the regression module of the hidden variables?

-  It is already common practice to use a buffer of past observations as input for the policy in locomotion applications [1][2]. However, the size of the buffer varies depending on the setting. How does the performance of your method change as the number of past observations increases?

   - [1] Joonho Lee et al., Learning quadrupedal locomotion over challenging terrain.
   - [2] Gabriel Margolis et al., Walk These Ways: Tuning Robot Control for Generalization with Multiplicity of Behavior

---

> ### Author Response · Authors · 2024-11-22
> **Response to Reviewer Amo3**
>
> Thank you for your review and comments. To respond/comment on your listed weaknesses:
> # Weaknesses:
> W1: While hardware support was indispensable for implementation, the primary focus of this paper remains on algorithmic contributions.Our algorithmic framework is designed to be flexible and extensible. For example, it can integrate multimodal inputs, like vision, tactile, and auditory data. The Transformer's ability to process diverse inputs offers a promising path to enhance robotic systems.\
> W2: The explanations for some of the questions are as follows: \
> $\bullet$ Regarding the ‘understanding of the interaction process with the environment,’ we have provided a detailed explanation in $\color{red}{Section\ 3.1.2\ Dynamical\ Environment\ Understanding}$. In this section, we define the concept of the interaction process between the robot and the environment, highlighting its highly dynamic nature and strong temporal correlations.
> As for ‘comprehending interaction information during dynamic prediction,’ we thoroughly discuss this in $\color{red}{Section\ 3.1.3 Dynamic\ Prediction}$, where we emphasize that dynamic prediction is an integral part of understanding the interaction process. During dynamic prediction, the robot leverages interaction information to ‘imagine’ the complete states of the physical world and itself resulting from each possible action. This capability allows the robot to proactively anticipate future dynamic changes and evaluate the potential value of each action, thereby enhancing its adaptability to diverse scenarios, especially in complex and unpredictable environments. Furthermore, this process strengthens the generalization of the robot's walking capabilities.
> We have revised the content in $\color{red}{Section\ 3.1.3\ Dynamic\ Prediction}$ to make this clearer to understand. \
> $\bullet$ Partial observations can only capture local information and fail to comprehensively represent the full complexity of the environment, making them insufficient to support the robot's decision-making requirements in complex environments. Due to these limitations, state estimators have been adopted[1][2]. Additionally, [3] such as teacher-student distillation are used to incorporate privileged information, addressing the constraints of partial observations. \
> $\bullet$ We assume that the critic has access to the full observation of the environment and accumulates historical observation information through hidden variables. In the actor network, we use Transformer to encode partial observations into a hidden variable sequence ${z}_{t}$. Since the critic’s hidden variables contain richer interaction information, we allow the critic’s hidden variables to supervise the learning of the actor’s hidden variables, thereby expanding the actor’s observation space. This mechanism enhances the actor’s performance in partially observable environments, enabling it to better capture environmental interaction features. For further details, please refer to $\color{red}{Section\ 3.2.3.}$ \
> W3: We have carefully reviewed the manuscript and addressed all the mentioned typos and formatting issues, including missing spaces before citations, missing words, and broken references.
> # Questions:
> Q1: The order of updates in our approach is primarily determined by practical considerations rather than a strict theoretical rationale. In practice, we found that optimizing the policy first and then the regression module of the hidden variables yielded comparable results to other update orders.
> However, there are some potential benefits to this specific order: \
> Policy Adaptation First: By optimizing the policy first, we ensure that it is well-aligned with the current dynamics of the environment. This alignment provides a solid basis for the subsequent optimization of the regression module. \
> Stable Latent Representations: Delaying the regression module's optimization slightly until after the policy update reduces the risk of destabilizing the latent space early in training. \
> We will explore this aspect in future work to better understand the implications of update order on performance and stability. \
> Q2: As discussed in the manuscript $\color{red}{Fig. 11\ Appendix\ A.3}$, we conducted experiments to analyze the impact of varying the history lengthon the performance of our method. The results show that time window length determines the model's context range in sequential tasks. Longer windows capture more dependencies but increase costs. In our experiments, lengths of 16 and 8 perform similarly and significantly better than others.
> # Reference
> [1] I Made Aswin Nahrendra et al., DreamWaQ: Learning Robust Quadrupedal Locomotion With Implicit Terrain Imagination via Deep Reinforcement Learning. ICRA 2023. \
> [2] Cui et al., Adapting Humanoid Locomotion over Challenging Terrain via Two-Phase Training. CORL 2024. \
> [3] Kumar et al., RMA: Rapid Motor Adaptation for Legged Robots. RSS 2021.

---

> ### Comment · Area_Chair_UsDo · 2024-11-25
>
> Dear Reviewer,
>
> Please provide feedback to the authors before the end of the discussion period, and in case of additional concerns, give them a chance to respond.
>
> Timeline: As a reminder, the review timeline is as follows:
>
> November 26: Last day for reviewers to ask questions to authors.
>
> November 27: Last day for authors to respond to reviewers.

---

> > ### Comment · Reviewer_Amo3 · 2024-11-28
> > **Response.**
> >
> > I appreciate the authors' effort in revising the paper,  fixing typos, and explaining the importance of the order of updates. I agree that the method is promising and have updated the score to 6.
> >
> > Nonetheless, I still have concerns about the clarity of multiple statements. In general, the task description (3.1.2) introduces vague and perhaps unnecessary definitions for "Dynamical Environment Understanding" and "Dynamic Prediction."  For instance,  L220 of the revised paper states, "Developing the robot’s cognitive ability to interact physically with a dynamic environment is crucial." How do you define a robot's cognitive ability?
> >
> > Furthermore, I stand by the statement that this is a good systems paper. While I agree that the proposed method could be flexible and extensible, its applicability was not showcased for other scenarios or applications, and that is out of the scope of the rebuttal period, which prevents me from raising the score any higher.

---

> > > ### Author Response · Authors · 2024-12-02
> > > **Response to Reviewer Amo3**
> > >
> > > Thank you for your thoughtful feedback and for updating the score. We appreciate your recognition of the revisions and the effort we’ve put into improving the clarity of the paper. We will now address your concerns regarding the clarity of certain statements.
> > >
> > > 1. Clarification of "Dynamical Environment Understanding" and "Dynamic Prediction":
> > >
> > > We understand that some of the terminology used, such as "Dynamical Environment Understanding" and "Dynamic Prediction," may seem vague. We now explain that "Dynamical Environment Understanding" refers to the model’s ability to recognize and adapt to changes in the environment, while "Dynamic Prediction" specifically involves forecasting the evolution of environmental states that affect the robot's motion.
> > >
> > > 2. Cognitive Ability of the Robot:
> > >
> > > Regarding your concern about the phrase "Developing the robot’s cognitive ability to interact physically with a dynamic environment," we agree that the definition of "cognitive ability" needs clarification. In our work, we use "cognitive ability" to refer to the robot’s capacity to perceive and process information about its environment and adjust its actions based on this understanding. This is central to our approach, enabling the robot to adapt to and interact with dynamic environmental factors.
> > >
> > > 3. Applicability and Generalization:
> > >
> > > We appreciate your recognition of the system's potential flexibility and extensibility. While we agree that showcasing the applicability to other scenarios would strengthen the paper, as you pointed out, this was outside the scope of the current work. However, we plan to investigate broader applications in future work and hope to provide additional insights in subsequent research.
> > >
> > > Once again, thank you for your constructive comments.

---

### Official Review · Reviewer_qhxP · 2024-11-04

**Soundness:** 2
**Presentation:** 1
**Contribution:** 1
**Rating:** 3
**Confidence:** 3

**Summary:**

The paper presents a framework for humanoid robot locomotion that uses the Transformer-XL architecture to learn a blind locomotion controller for a range of different, flat and non-flat, terrains. The key contribution is the introduction of a physical interaction world model that "understands" the dynamic interaction between the robot and its environment. The model is trained in simulation and deployed both in sum and on a real-world robot. The authors presents a set of comparisons with other methods and a baseline approach. The results show how the proposed framework is able to walk over a set of trial terrains.

**Strengths:**

The results in simulation show that the approach outperforms the baseline and related methods, and the oracle-based approach in vehicle tracking.
The combination of the actor-critic with the transformer-xl approaches and the application to humanoid locomotion is somewhat novel in a blind locomotion context.
The deployment on a real robot platform is an added strength for the work.

**Weaknesses:**

The main weakness of the paper is the lack of robust evaluation especially in the case of the real robot examples. Traditionally for locomotion controllers one can use measures as robustness to disturbances and noise to evaluate the robustness of a given locomotion approach, this would help in quantifying the resulting approach.

Why is the baseline chosen a good representation of what is available as state of the art for humanoid locomotion?

Parts of the work are not presented in good detail and there are parts of the text that are unclear or need to be expanded on. For example, line 233, what is sufficient information? in line 250, "The actor hidden variables are guided by the critic hidden variables to learn more interaction information" this is unclear, in line 260, what is the contact mask and what are the clock inputs? lines 275 and 286 what do the critic_trans(.) and actor_trans(.) functions do?

The conclusion section is hard to follow, especially where many different vague terms are used in a single sentence, for example, "This framework incorporates the world model concept, forming self-awareness, environmental understanding, dynamic prediction, and observation expansion through historical sequences". It is not clear how self-awareness or environmental understanding has been addressed with the approach.

Line 071. Sentence is not clear.
Line 088. What does affordance-based mean in this context?
Related work section, many spaces are missing.
Line 261, system(\theta_{xy}) is not defined
Line 317, Better capture with respect to what?

**Questions:**

The computational cost of the approach is not discussed in the paper, a discussion on training and deployment could add to the presentation of the approach.

Line 421, The method outperforms the oracle-based method that has access to privileged information. A discussion to how this is possible or what information is being exploited to achieve this can add to the depth of the analysis of the results.

Latent layer analysis, it is not clear what this represents. What do we expect to see here and how the plots support this?

Lin 470, it is unclear what self-awareness and environmental perception refers to.

---

> ### Author Response · Authors · 2024-11-22
> **Response to Reviewer qhxP**
>
> Thank you for your review and comments. To respond/comment on your listed weaknesses:
> # Weakness
> W1: We applied external forces to the robot's legs in real-world environments to simulate disturbances. $\color{red}{Fig. 9}$ The experiments demonstrated that our method enables the robot to swiftly adapt to changes, adjust its gait, and maintain balance under external disturbances. Detailed results can be found in the supplementary video. \
> W2: The baseline method in our work is the classical PPO-based RL algorithm, which is widely adopted in both recent humanoid locomotion studies [1][2][3] and classical legged robot RL research. It is commonly used as a benchmark for performance comparison in these fields, demonstrating its relevance and validity as a representative baseline. We have included comparisons with other advanced architectures. $\color{red}{Fig. 4}$ \
> W3: \
> $\bullet$ “Sufficient Information” is presented in contrast to "partial observations" mentioned earlier in the text. Sufficient information additionally includes environmental data (e.g., friction coefficients) and interaction information between the robot and the environment. $\color{red}{Line\ 234}$ \
> $\bullet$ In $\color{red}{Line\ 249}$, this refers to the mechanism where the actor network leverages the hidden variables from the critic network during training. Since the critic has access to privileged information (asymmetric setup), its hidden variables encode richer contextual data. These variables serve as supervision signals for the actor network, helping it learn better representations of the environment-robot interactions. More details can be found in  $\color{red}{Section\ 3.2.3}$ \
> $\bullet$ “Contact Mask”: This indicates the contact states of the robot's feet with the ground; “Clock Inputs”: Clock input refers to a periodic signal input used in the reinforcement learning framework to provide temporal information. $\color{red}{Line\ 260}$ \
> $\bullet$ critic_trans: Encodes the complete observation space, including privileged information, to generate contextualized features for the critic; actor_trans: Processes the partial observation data, utilizing temporal dependencies to produce robust policy outputs.  tips:we refer to these two functions as $f_{\psi}$. $\color{red}{Section\ 3.2.2\  3.2.3}$ \
> W4: has been updated. $\color{red}{Conclusion}$ \
> W5: \
> $\bullet$ Change to "Environmental information and robotic motion information originate from inherently different domains." $\color{red}{Line\ 72}$ \
> $\bullet$ "Affordance-Based" is Similar to usable. $\color{red}{Line\ 90}$ \
> $\bullet$ Space added. $\color{red}{Section\ 2}$ \
> $\bullet$ Change to "θ_xyz". $\color{red}{Line\ 262}$ \
> $\bullet$ "Better capture" refers to the ability of our method to better capture the interaction dynamics between the robot and its environment compared to other baselines. This comparison highlights the effectiveness of our approach in modeling these interactions.  Tips: We have revised this section and removed this term. $\color{red}{Section\ 3.2.5\ Reconstruction\ Loss }$
> # Question
> Q1: \
> Training Cost: The training process was conducted on a machine equipped with an NVIDIA RTX V100 GPU. (40G) Training typically converges within 2 days for our standard configurations. \
> Deployment Cost: For real-world deployment, the inference runs efficiently on an NVIDIA Jetson NX, a lightweight edge device. The Transformer-XL model achieves real-time performance (100 Hz control loop) without requiring additional computational resources. \
> Q2: In our work, the Oracle method is designed as a baseline that has access to privileged information, including proprioceptive data and privileged data. This baseline employs an MLP architecture to process the combined input of proprioceptive and privileged information. Our method, in contrast, which introduces several advantages: Temporal Modeling Capability、Implicit Environmental Interaction Modeling、Dynamic Prediction and Observation Expansion \
> Q3: We referred to the experiments in [4]. We selected the outputs of 6 neurons from the hidden layer, which has a total of 128 dimensions, for visualization. The visualization shows changes in the hidden layer responses as the robot moves from plane to an incline and back to plane. These changes intuitively demonstrate how the robot adapts to different terrains, reflecting the network’s ability to recognize and respond to terrain changes, as well as the real-time adjustment of its control strategy. \
> Q4: Refer to W4.
> # Reference
> [1] Gu et al., Advancing Humanoid Locomotion: Mastering Challenging Terrains with Denoising World Model Learning, RSS 2024 \
> [2] Cui et al., Adapting Humanoid Locomotion over Challenging Terrain via Two-Phase Training.Corl 2024 \
> [3] Marum et al., Learning Perceptive Bipedal Locomotion over Irregular Terrain. ICRA 2024 \
> [4] Ilija Radosavovic et al., Real-world humanoid locomotion with reinforcement learning.Sciencerobotic 2024

---

> ### Comment · Area_Chair_UsDo · 2024-11-25
>
> Dear Reviewer,
>
> Please provide feedback to the authors before the end of the discussion period, and in case of additional concerns, give them a chance to respond.
>
> Timeline: As a reminder, the review timeline is as follows:
>
> November 26: Last day for reviewers to ask questions to authors.
>
> November 27: Last day for authors to respond to reviewers.

---

> ### Author Response · Authors · 2024-11-28
> **Response to Reviewer qhxP**
>
> # Weaknesses
> > W2：Why is the baseline chosen a good representation of what is available as state of the art for humanoid locomotion?
>
> In our previous response, we provided an explanation for why the baseline was chosen. To further address the reviewer's suggestion, we have added an additional experiment comparing our approach with the method proposed in [1]. This comparison was made based on the feedback from another reviewer, who pointed out that [1] demonstrated superior performance and suggested it would be valuable for us to include a comparison.
>
> Specifically, [1] proposes Denoising World Model Learning, while our approach employs a fundamentally different world model. In the revised manuscript, we have added comparative experiments in simulation to evaluate the terrain level progression curves of both methods. $\color{red}{Fig. 13}$ demonstrates that our method achieves a faster progression in terrain complexity.
>
> Tips："terrain level" refers to the complexity of the terrain, and it is determined by assigning a level to different terrain attributes, such as the height of steps. For example, in a "stair up" environment, stair heights ranging from 0.5 cm to 1.2 cm are divided into different levels, with 0.5 cm being level 0 and 1.2 cm being level 6.
>
> > W4: The conclusion section is hard to follow, especially where many different vague terms are used in a single sentence, for example, "This framework incorporates the world model concept, forming self-awareness, environmental understanding, dynamic prediction, and observation expansion through historical sequences". It is not clear how self-awareness or environmental understanding has been addressed with the approach.
>
> Thank you for your valuable feedback on our conclusion section. We appreciate your point about the clarity of certain terms. We realize that some of the terminology used, such as "self-awareness" and "environmental understanding," was not the most appropriate. To address this, we have revised the conclusion to provide a clearer explanation of the framework and its contributions.
>
> The updated conclusion now reads:
>
> "In this work, we propose Interactive World Models for Humanoid Robots (HuWo), a novel framework designed to tackle the challenges of humanoid robot locomotion in complex environments. Our framework leverages the world model concept within an asymmetric actor-critic structure, where the hidden layers of Transformer-XL enable end-to-end implicit modeling of dynamic interactions between the robot and its environment. This approach facilitates robust decision-making by expanding the observation space through historical sequences and dynamic predictions.
>
> The effectiveness of our method is demonstrated through experiments showing robust walking performance in diverse real-world environments, including zero-shot sim-to-real transfer. These results highlight our framework’s ability to model environmental dynamics and adapt to changing conditions. Future work will focus on extending this framework to achieve full-body coordination, including free arm movement, for more versatile and natural locomotion."
>
> We hope that this revised version makes the explanation clearer and better reflects the contributions of our approach. We look forward to your further comments.
>
> # Reference
> [1] Gu et al., Advancing Humanoid Locomotion: Mastering Challenging Terrains with Denoising World Model Learning, RSS 2024

---

> > ### Author Response · Authors · 2024-11-28
> > **Response to Reviewer qhxP**
> >
> > Dear Reviewer qhxP,
> >
> > First and foremost, we sincerely thank you for your valuable feedback on our paper. Based on your suggestions, we have made several revisions, which are summarized as follows:
> >
> > $\bullet$ We have corrected typographical errors and unclear sentences.
> >
> > $\bullet$ We have provided clarifications for certain terms and experimental details, and have modified sections that could lead to misunderstandings.
> >
> > $\bullet$ We have added additional experimental comparisons with the latest methods.
> >
> > $\bullet$ We have included descriptions of the training and deployment costs.
> >
> > Once again, thank you for your time and effort in reviewing our work. We look forward to your feedback.
> >
> > Best regards,
> >
> > Authors of 13357

---

### Author Response · Authors · 2024-11-26
**Further Discussion**

Dear Reviewers,

We sincerely appreciate your thoughtful comments and valuable suggestions. As the final revision deadline is approaching, we would like to kindly ask if our responses have adequately addressed your concerns. We remain open to further discussion or clarification and welcome any additional feedback to improve our work before the submission deadline.

Thank you again for your time and insights.

Best regards,
Authors of 13357

---

### Meta-Review · Area_Chair_UsDo · 2024-12-20

**Metareview:**

This paper presents a framework for humanoid robot locomotion using a Transformer-XL-based reinforcement learning approach to model physical interaction dynamics. While the method demonstrates robustness in simulation and some real-world scenarios, the reviewers raised several critical concerns regarding the methodological contributions of the paper and the clarity in the presentation.

The reviewers acknowledged the promise of the approach, particularly in addressing challenging locomotion tasks and the integration of world models with reinforcement learning for solving the task at hand. However, concerns were raised about the limited quantitative evaluation of real-world performance, the lack of comprehensive ablation studies, and vague or unclear descriptions of key components. Furthermore, the application focus of the methodology on humanoid locomotion makes the paper feel more suitable for a robotics conference rather than ICLR, which typically prioritizes broader machine learning contributions.

During the rebuttal phase, the authors addressed some issues by revising sections for clarity, adding baseline comparisons, and responding to specific technical concerns. However, these revisions were insufficient to resolve core weaknesses, including the limited exploration of generalization across domains and a lack of rigorous real-world testing.

In the view of the AC, this is a good systems contribution with potential applications in humanoid locomotion, but it does not meet the standards of novelty for ICLR, and seems to be better suited for a robotics venue.

**Additional Comments On Reviewer Discussion:**

The authors provided detailed responses addressing the reviewers’ concerns during the rebuttal period. While their efforts clarified some aspects, key issues remained unresolved, leaving the rebuttal partially convincing. Below, I outline the major points raised by reviewers and how they were addressed by the authors, referencing specific reviewers where applicable.

1.	Relevance to ICLR and Generalization

- Reviewer Amo3 noted that the paper’s domain-specific focus on humanoid locomotion made it feel better suited for a robotics venue: “The scope of the paper is very specific to one domain, and while the proposed method is potentially useful for other applications, showcasing its usefulness in different domains will require further experiments on a broader set of environments and tasks.”

- Author Response: The authors emphasized their method’s flexibility for multimodal inputs (e.g., vision and tactile) and their intent to explore broader applications in future work. However, no experimental evidence was provided, which left this point inadequately addressed.

2.	Evaluation and Ablation Studies

- Reviewer QaRB highlighted missing ablations: “The ablation on the dynamics estimation module is missing. There is also no ablation on the latent variable regression. These ablations are mentioned in the text but are not in the figures.”

- Author Response: The authors clarified that these ablations were included in the appendix but they were not well-referenced in the main text. They updated the manuscript to address this oversight, but reviewers remained unsatisfied with the depth of evaluation.

3.	Baseline Comparisons:

- Reviewer AAFX raised concerns about inadequate baselines, noting: “Could you show more evidence that the proposed method is superior to baseline methods in the real world? Have you considered Gu et al. as a baseline?”

- Author Response: The authors included a new comparison with Gu et al.’s method (RSS 2024) in simulation, showing faster terrain adaptation. While appreciated, reviewers (e.g., Amo3) felt this addition was insufficient to resolve baseline concerns.

4.	Clarity and Presentation:

-	Reviewer qhxP noted vague terminology, such as “self-awareness” and “dynamic prediction”, stating: “It is not clear how self-awareness or environmental understanding has been addressed with the approach.”

-	Author Response: The authors revised the paper to clarify these terms and removed overly broad claims. They also improved figure readability and fixed typographical errors. These changes were generally well-received, although Reviewer Amo3 remarked that the revised explanations still lacked depth.

5.	Sim-to-Real Gap and Real-World Validation
- Reviewer AAFX critiqued the limited real-world testing, stating: “The evaluation would be more robust with additional quantitative results from real-world experiments. The sim-to-real gap can be significant.”

-    Author Response: The authors argued that their simulation environment incorporated domain randomization and curriculum learning to reduce the sim-to-real gap. They defended the lack of quantitative real-world results, citing safety constraints. While Reviewer Amo3 acknowledged these challenges, they maintained that stronger real-world evidence was needed.


Overall, the AC judges the paper as borderline and considers that is better suited to robotics-related venues.

---

### Decision · Program_Chairs · 2025-01-22

Reject